# Natural Deep Eutectic Solvents for the Extraction of Bioactive Steroidal Saponins from Dioscoreae Nipponicae Rhizoma

**DOI:** 10.3390/molecules26072079

**Published:** 2021-04-05

**Authors:** Gui-Ya Yang, Jun-Na Song, Ya-Qing Chang, Lei Wang, Yu-Guang Zheng, Dan Zhang, Long Guo

**Affiliations:** 1Traditional Chinese Medicine Processing Technology Innovation Center of Hebei Province, Hebei University of Chinese Medicine, Shijiazhuang 050200, China; Yangguiya_310@163.com (G.-Y.Y.); junnassong@163.com (J.-N.S.); Czuoya_123@163.com (Y.-Q.C.); wanglei1031@126.com (L.W.); zyg314@163.com (Y.-G.Z.); 2Hebei Chemical & Pharmaceutical College, Shijiazhuang 050200, China

**Keywords:** natural deep eutectic solvents, Dioscoreae Nipponicae Rhizoma, steroidal saponins, response surface methodology, extraction

## Abstract

In the present study, a simple and environmentally friendly extraction method based on natural deep eutectic solvents (NADESs) was established to extract four bioactive steroidal saponins from Dioscoreae Nipponicae Rhizoma (DNR). A total of twenty-one types of choline chloride, betaine, and L-proline based NADESs were tailored, and the NADES composed of 1:1 molar ratio of choline chloride and malonic acid showed the best extraction efficiency for the four steroidal saponins compared with other NADESs. Then, the extraction parameters for extraction of steroidal saponins by selected tailor-made NADES were optimized using response surface methodology and the optimal extraction conditions are extraction time, 23.5 min; liquid–solid ratio, 57.5 mL/g; and water content, 54%. The microstructure of the DNR powder before and after ultrasonic extraction by conventional solvents (water and methanol) and the selected NADES were observed using field emission scanning electron microscope. In addition, the four steroidal saponins were recovered from NADESs by D101 macroporous resin with a satisfactory recovery yield between 67.27% and 79.90%. The present research demonstrates that NADESs are a suitable green media for the extraction of the bioactive steroidal saponins from DNR, and have a great potential as possible alternatives to organic solvents for efficiently extracting bioactive compounds from natural products.

## 1. Introduction

Dioscoreae Nipponicae Rhizoma (DNR), the dried rhizome of *Dioscorea nipponica* Makino, is widely distributed in China, South Korea, and Russia, and has been used as a traditional herbal medicine for relieving cough and asthma, eliminating rheumatic aches, alleviating pain, and improving blood circulation [1]. Modern pharmacological studies have discovered that DNR possesses anti-tumor, anti-inflammatory, anti-diuretic, analgesic, anti-tussive, immunoregulatory, and cardiovascular properties [2,3,4,5,6]. Phytochemical studies disclosed that steroidal saponins, such as protodioscin, protogracillin, pseudoprotodioscin, and pseudoprotogracillin are the major bioactive constituents in DNR, which are mainly responsible for most of the pharmacological effects of this herbal medicine [7,8]. Therefore, the content of the bioactive steroidal saponins is an important index for the quality control and clinical practice of DNR. Several analytical methods, including thin layer chromatography, high performance liquid chromatography, and high performance liquid chromatography coupled with mass spectrometry, have been used for qualitative and quantitate analysis of bioactive steroidal saponins in DNR [9,10]. However, research focused on the extraction method for the bioactive steroidal saponins from DNR are still limited.

Conventional organic solvents, such as methanol, ethyl acetate, acetone, and chloroform, are commonly used in the extraction of bioactive components from natural sources. However, the consumption of large amounts of these volatile and hazardous organic solvents may contribute to environmental pollution and leave unacceptable solvent residues in extracts. Green extraction methods which are environmentally friendly and sustainable for sample preparation have received more and more attention in past years [11].

Since being introduced as a new type of green solvents, deep eutectic solvents (DESs) have rapidly gained great interest as sustainable alternatives to hazardous organic solvents [12]. DESs are composed of a mixture consisting of hydrogen bond acceptors (HBAs) with hydrogen bond donors (HBDs), which are able to form intermolecular hydrogen bonds and van der Waals interactions. Owing to their superior properties, such as negligible volatility, non-flammability, adjustable viscosity, cost-effectiveness, and facile production processes, DESs have been extensively used in many different fields such as pharmaceuticals, catalysis, electrochemistry, and extractions [13,14,15,16,17]. When DESs were prepared by mixing two or more naturally occurring and biodegradable components, they were called natural deep eutectic solvents (NADESs). The availability, low cost, biodegradability, and environmental friendliness of the components makes the NADESs versatile alternatives to conventional organic solvents [18]. Recently, several studied have used NADESs for extraction and separation of different types of bioactive compounds, such as phenolic acids, flavonoids, and alkaloids from various plant materials [19,20,21]. Nonetheless, the number of reports on the application of NADESs for the extraction of bioactive compounds is still limited, and the efficiency of NADESs on extraction of bioactive steroidal saponins from DNR still remains unknown.

In the present study, in order to evaluate NADESs for the extraction of steroidal saponins, a series of choline chloride (ChCl), betaine (Bet), and l-proline (Pro) based NADESs were tailored for ultrasonic extraction of four bioactive steroidal saponins, including protodioscin, protogracillin, pseudoprotodioscin, and pseudoprotogracillin from DNR. Then, the extraction conditions for the four steroidal saponins, including the water content of NADESs, extraction time, and liquid–solid ratio were systematically optimized using response surface methodology (RSM), and the microstructure of the raw and extracted DNR powder were observed by field emission scanning electron microscope. Moreover, the recovery of the four steroidal saponins from tailored NADESs was tentatively carried out by D101 macroporous resins.

## 2. Results and Discussion

### 2.1. HPLC Conditions and Method Validation

In order to achieve a rapid and efficient analysis of the four bioactive steroidal saponins (protodioscin, protogracillin, pseudoprotodioscin, and pseudoprotogracillin) in DNR, several HPLC conditions, including mobile phases (water-methanol, water–acetonitrile, formic acid water–methanol, and formic acid water–acetonitrile), flow rates (0.9, 1.0, and 1.1 mL/min), and column temperatures (20, 30, and 40 °C) were compared and optimized. As a result, the water–acetonitrile system at 30 °C with a flow rate of 1.0 mL/min was selected for the suitable analysis duration, greater separation ability, and better peak shapes of the four analytes. The chromatographic peaks of: protodioscin (1), protogracillin (2), pseudoprotodioscin (3), and pseudoprotogracillin (4) were confirmed by comparing their retention times with the reference standards. The typical HPLC chromatograms of DNR sample and the four steroidal saponins reference standards are shown in Figure 1.

Method validation of quantitative analysis of the four bioactive steroidal saponins in DNR was performed. The linearity, limit of detections (LODs), limit of quantifications (LOQs), precision, repeatability, stability, and accuracy were validated. The calibration curves of the four steroidal saponins were performed with six different concentrations of standard solutions in triplicate. All calibration curves were of good linearity with high correlation coefficient (*r*^2^ > 0.9992) over the tested range. The LODs and LOQs of the four analytes were defined by the concentration that generated peaks with signal-to-noise values of 3 and 10 using standard solutions. The precision of the method was determined by the intra- and inter-day variations. For intra-day tests, the same sample was analyzed six times within the same day, while for inter-day tests, the sample was examined in duplicates for three consecutive days. The relative standard deviations (RSDs) of intra- and inter-day precisions were less than 1.4% and 2.7%, respectively. For the repeatability test, six replicates of the same sample were prepared and analyzed, and for the stability test, the same sample was stored at room temperature and analyzed by replicate injection analysis at 0, 2, 4, 8, 12, and 24 h. The repeatability presented as RSDs were less than 2.1%, and the stability were less than 3.0%. The recovery test was used to evaluate the accuracy of the method. A known amount of the four steroidal saponins reference standard solutions were added into the same samples in sextuplicate, and then the samples were extracted and analyzed with the same procedures as described in Section 3.3. The recovery of each analyte was calculated by the equation: Recovery (%) = (Detected amount − Original amount)/Spiked amount × 100%. The overall recoveries of the four analytes were in the range of 98.8–107.5% with RSDs less than 2.3%. All of the above data are shown in Table 1, which indicates the established method is precise, accurate, and sensitive for the quantification of the four steroidal saponins in DNR samples.

### 2.2. Preparation of NADESs

In general, the extraction efficiency for bioactive compounds from plant sources could potentially be influenced by both the extraction conditions and the physicochemical properties of the extraction solvents. Similarly, NADESs are composed of a mixture consisting of HBAs and HBDs of natural origins, and the chemical compositions of NADESs determines their physicochemical properties and consequently greatly influences extraction efficiency of bioactive compounds from natural products. In this work, with the aim to select the most appropriate NADES for the extraction of bioactive steroidal saponins, three groups of HBDs (alcohols, carboxylic acid, and amides) were used in combinations with three types of HBAs (choline chloride, betaine, and L-PROLINE) to prepare NADESs. A total of thirty combinations of NADESs were initially prepared, and twenty-one of them were eventually found to be stable as a clear, viscous liquid without precipitation over the course of time (Table 2).

During the preparation of different types of NADESs, we found that the chemical structures of HBAs play essential roles in the formation and stability of NADESs. NADESs could be successfully formed by heating ChCl or Bet with lactic acid or malonic acid, in contrast, Pro was proved incapable of producing a stable liquid when heated with lactic acid or malonic acid. Amides-based NADESs could be prepared when the HBAs was ChCl, but when the HBAs were Bet and Pro, amides-based NADESs could not successfully be formed. When HBDs were in a liquid form, such as glycerol and ethylene glycol, it was easier to synthesize a clear and less viscous NADESs, and the carboxylic acid-based NADESs were also easier to synthesize. In addition, the different ratios of HBAs and HBDs might affect the synthesis and stability of NADESs. For example, when ChCl were heated with malonic acid in the mole ratios of 2:1, 1.5:1, 1:1, 1:1.5, and 1:2, a clear liquid can be only prepared with 1.5:1, 1:1, or 1:1.5 mole ratio.

### 2.3. Screening of NADESs for Extraction of Bioactive Steroidal Saponins

Compared with conventional extraction solvents, the high viscosity of NADESs restricted their application in extraction. High viscosity may impair the efficiency of extraction as slow mass transfer occurs, and the low viscosity of the NADESs led to high diffusivity and hence improved the extraction performance [22]. In the initial screening experiments, extraction conditions were adjusted to reduce the viscosity by adding a certain amount of water. It is reported that a large excess of water could break the hydrogen bonds between the NADESs components and lose the eutectic property of the solvents produced [23]. Therefore, the NADESs were diluted to 70% (*v*/*v*) with water to decrease viscosities for enhancing the extraction efficiency.

Extraction yields of the four steroidal saponins (protodioscin, protogracillin, pseudoprotodioscin, and pseudoprotogracillin) and the total four steroidal saponins with different types of NADESs are shown in Figure 2. The results of initial screening experiments indicated that the extraction efficiency for the four bioactive steroidal saponins was influenced by the types of NADESs solvents, and different types of DESs resulted in different extraction yields. In general, extraction yields of the four steroidal saponins followed the order protodioscin > protogracillin > pseudoprotodioscin > pseudoprotogracillin. It was obvious that protodioscin showed higher extraction efficiency in ChCl-Lac, followed by ChCl-Maa, ChCl-Mu, and Bet-Lac. Similarly, protogracillin exhibited higher extraction efficiency in ChCl-Lac, ChCl-Maa, ChCl-Mu, and Bet-Lac. For pseudoprotodioscin and pseudoprotogracillin, NADESs ChCl-Mal, ChCl-Maa, and Pro-Xyl led to higher extraction yields. The total extraction yields of the four steroidal saponins were also calculated, and ChCl-Mal and ChCl-Lac showed relative higher extraction efficiency for the four bioactive steroidal saponins compared with other NADESs.

To get insight into the NADESs’ excellence in the extraction of bioactive steroidal saponins from DNR, the extraction yields of different NADESs and conventional efficient solvents (methanol and water) were compared. For protodioscin and protogracillin, almost all the NADESs exhibited lower extraction efficiency compared with methanol, and the ChCl-Eg, ChCl-Lac, ChCl-Mal, ChCl-Mu, Bet-Lac, and Bet-Mal exhibited higher extraction efficiency compared with water. For pseudoprotodioscin and pseudoprotogracillin, the ChCl-Maa, ChCl-Mal, Pro-Xyl, and Pro-Ox exhibited higher extraction efficiency compared with methanol and water. For the total four steroidal saponins, the extraction efficiency of ChCl-Eg, ChCl-Lac, ChCl-Mal, ChCl-Mu, ChCl-Am, Bet-Lac, and Bet-Mal was higher than water with the concentration 59.20 ± 0.02, 64.27 ± 0.03, 64.99 ± 0.02, 61.36 ± 0.01, 55.31 ± 0.03, 60.05 ± 0.02, and 57.64 ± 0.01 mg/g, respectively. Compared to methanol, most NADESs showed lower extraction yields of the total four steroidal saponins, but ChCl-Lac (64.27 ± 0.03 mg/g) and ChCl-Mal (64.99 ± 0.02 mg/g) produced approximate extraction efficiency. Thus, ChCl-Mal was selected as the best NADES for extraction of steroidal saponins from DNR and applied in further tests based on the initial screening results.

### 2.4. Optimization of the Extraction Conditions for Bioactive Steroidal Saponins

#### 2.4.1. Optimization of Molar Ratios of ChCl-Mal

The molar ratio of HBAs and HBDs has been regarded as an important factor that influences the surface tension and polarity of NADESs [24], which could affect the extraction efficiency of target components from natural biomass. As described above, ChCl-Mal produced the highest extraction efficiency of steroidal saponins from DNR. The molar ratios of ChCl-Mal were further investigated. The eutectic mixtures of ChCl-Mal were prepared at mole ratios of 2:1, 1.5:1, 1:1, 1:1.5, and 1:2, and homogeneous and transparent liquid could be synthesized with 1.5:1, 1:1, or 1:1.5 mole ratios. Thus, the ChCl-Mal with the mole ratios 1.5:1, 1:1, and 1:1.5 were used to extract bioactive steroidal saponins from DNR. As shown in Figure 3, the results indicated that the different molar ratios of ChCl-Mal have impact on the extraction yields of protodioscin and protogracillin, but have no impact on pseudoprotodioscin and pseudoprotogracillin. The optimal molar ratio of ChCl-Mal was 1:1 and the extraction yields of protodioscin, protogracillin, pseudoprotodioscin, and pseudoprotogracillin were 29.30 ± 0.67, 15.86 ± 0.37, 9.71 ± 0.23, and 3.66 ± 0.07 mg/g, respectively.

#### 2.4.2. Optimization of the Extraction Conditions by Response Surface Methodology

RSM was an effective method to design and analyze tests with independent variables upon different levels to optimize the results, and had been widely used in optimization of extraction processes [25,26]. The above extraction investigations showcased that ChCl-Mal with the mole ratio 1:1 was the best NADES for extraction of bioactive steroidal saponins from DNR. In order to obtain the optimal extraction efficiency for steroidal saponins, several extraction conditions that could affect the extraction efficiencies were optimized by RSM. Similar to previous researches, three independent variables, including water content in NADESs, extraction time, and liquid–solid ratios were further optimized. After determining the range of extraction factors on the basis of preliminary single-factor tests, the water content in NADESs (A, 20–40%), extraction time (B, 10–40 min), and liquid–solid ratios (C, 10–50 mL/g) at three level (−1, 0, 1) in the selected NADES (ChCl-Mal) were evaluated using Box–Behnken design (BBD). The extraction yields of four bioactive steroidal saponins were set as the response of the design experiments. The experimental orders, levels of variables, and response values are summarized in Table 3.

The variables and response were processed to build a mathematical regression model for steroidal saponins. The model was expressed as second order polynomial quadratic equation for the extraction yields (Y) and coded variables (A, B, and C) as follows:Y = 63.27 + 3.18A + 3.07B + 2.77C − 3.44AB − 1.38AC − 2.26BC + 0.41A^2^ − 0.66B^2^ − 0.89C^2^(1)

The quality of regression model for steroidal saponins was evaluated in terms of the square of correlation coefficient (R^2^) and the Lack of Fit by the analysis of variance (ANOVA) at the 95% confidence level. As shown in Table 4, the R^2^ value was 0.9153, indicating that the experimental data were in relatively good agreement with predicted extraction yields. The Lack of Fit value was insignificant for the response with *p*-value of 0.4976 (*p* > 0.05), which indicated that the regression model fitted the models well and could suitably explain the data.

Statistical analysis and 3D response plots (Figure 4) obviously illustrated the significant variables affecting extraction yields of steroidal saponins and the interaction effects between the variables. It was clear that all the three variables, water content in NADESs, extraction time, and liquid–solid ratios showed significant effects on the extraction efficiency of steroidal saponins (*p* < 0.05). Based on the regression model, the optimum condition of ChCl-Mal for extraction of four bioactive steroidal saponins from DNR was as following: water content in NADESs, 39.69%; extraction time, 19.98 min; liquid–solid ratios, 32.28 mL/g. For the convenience of practical operation, the optimum condition was set at water content in NADESs, 40%; extraction time, 20 min; liquid–solid ratios, 32 mL/g. Triplicate experiments were carried out under the optimal extraction conditions and mean values of experimental results were compared with the predicted values. Under the optimum conditions, the actual extraction yield of the total four steroidal saponins was 66.82 mg/g, which was closed to the predicted extraction yield of 66.89 mg/g. This results indicated that the model was credible for optimization of extraction parameters of steroidal saponins from DNR.

### 2.5. Microstructure of Plant Material

Several previous studies have been reported that the NADESs could replace the organic solvents as extraction media for extraction of bioactive compounds from natural products due to their great extraction performance, which agrees with the results of this study [27,28]. However, the explanation of the superior extraction performance of some NADESs are still unclear. It is well known that the ultrasonic extraction could utilize the ultrasound irradiation energy to produce cavitation phenomenon, along with the formation of bubbles that collapsed in the multiple directions of the plant surface. This external force could break the plant cell wall and reduce the particle size, facilitating swelling, and extraction solvent permeation into the cells to release the bioactive compounds from the vacuole [29]. 

In the present study, the microstructures of the DNR powder before and after ultrasonic extraction in water, methanol, and the selected NADES (ChCl-Mal) were observed by field emission scanning electron microscope. As shown in Figure 5, the DNR powder before ultrasonic extraction displayed a rough and coarse surface with many flakes (Figure 5a), whereas the surface of powder after ultrasonic extraction in water, methanol, and ChCl-Mal displayed some breakages along the edges without any flakes (Figure 5b–d). The results illustrated that both the conventional solvents (water and methanol) and NADES (ChCl-Mal) could destroy the cells and cell walls as well as dissolved those weak flakes during ultrasonic extraction [30].

### 2.6. Recovery of Bioactive Steroidal Saponins from NADESs

The recovery of extracted bioactive compounds from NADESs is a challenge due to the high water miscibility and non-volatilization property of NADESs. The D101 macroporous resin is a common, accessible, and inexpensive packing material for column chromatography. In this study, the recovery of the four bioactive steroidal saponins from ChCl-Mal was tentatively applied by using D101 macroporous resin column chromatography according to our previous research [31]. As described in Section 3.8, the recoveries of the four bioactive steroidal saponins, protodioscin, protogracillin, pseudoprotodioscin, and pseudoprotogracillin were 79.90%, 68.12%, 67.27%, and 74.8%, respectively. The results indicated that D101 macroporous resin could effectively adsorb steroidal saponins while the polar ingredients of NADESs could be elected with water, and most steroidal saponins could be obtained with ethanol after eluting the polar compounds with water. In summary, the recovery of the four steroidal saponins in NADESs extraction solution could be conveniently, readily, and efficiently achieved using D101 microporous resin.

## 3. Materials and Methods

### 3.1. Materials and Reagents

DNR samples were purchased from a local Traditional Chinese Medicine market (Anguo, Hebei, China). The samples were identified by Associate Professor Long Guo, and the voucher specimens have been deposited in Hebei University of Chinese Medicine, Shijiazhuang, China. The DNR samples were dried in the shade and powered by a disintegrator, then sieved through 60 mesh and stored in the desiccator before use.

Choline chloride (ChCl), betaine (Bet), L-proline (Pro), ethylene glycol (Eg), glycerol (Gly), xylitol (Xyl), D-sorbitol (Sor), lactic acid (Lac), malonic acid (Mal), DL-malic acid (Maa), Oxalic acid (Ox), Citrate acid (Ca), l-Methylurea (Mu), Acetamide (Am), and Urea (Ur) were purchased from Aladdin Biochemical Technology (Shanghai, China). The reference compounds of protodioscin, protogracillin, pseudoprotodioscin, and pseudoprotogracillin (purities > 95%) were obtained from Chengdu Biopurify Phytochemicals Company (Chengdu, China). The chemical structures of the four steroidal saponins are shown in Figure 6. Acetonitrile and methanol (HPLC grade) were purchased from Fisher Scientific (Geel, Belgium). Deionized water was prepared by a Synergy water purification system (Millipore, Billerica, USA). Other reagents and chemicals were of analytical grade.

### 3.2. Preparation of NADESs

NADESs can be synthesized by a heating method according to the previous study [31]. Briefly, DESs were synthesized by simply mixing the hydrogen bond acceptors (HBAs) and hydrogen bond donors (HBDs) at a proper molar ratio with magnetic agitation at 80.0 °C until a homogeneous and transparent liquid formed.

### 3.3. Extraction Procedures

For initial screening of NADESs for extraction of the four bioactive steroidal saponins from DNR, an accurately weighed DNR powder (50 mg) was extracted with 1 mL of prepared NADESs containing 30% water in a 2-mL centrifuge tube by ultrasonic (300 W, 40 kHz) for 20 min at room temperature. After extraction, the mixed suspension was centrifuged at 13,000 rpm/min for 10 min. Then, the supernatant was diluted eight times using 50% methanol and filtered through membrane filters (0.22 μm) prior to HPLC analysis. Each extraction was performed in triplicate.

### 3.4. Preparation of Standard Solutions

The reference compounds of protodioscin, protogracillin, pseudoprotodioscin, and pseudoprotogracillin were accurately weighed and dissolved in DES (ChCl-Mal), and then diluted eight times with 50% methanol to the concentrations of 840, 490, 640, and 320 μg/mL, respectively. The standard solutions containing four reference standards were prepared by appropriate dilution to a series of proper concentrations. The standard solutions were stored at 4 °C for further HPLC analysis.

### 3.5. Determination of Bioactive Steroidal Saponins by HPLC

The concentrations of four steroidal saponins extracted by NADESs were determined using an Agilent 1260 HPLC system (Agilent, San Jose, CA, USA) equipped with a quaternary pump, an autosampler, a thermostatic column compartment and a diode array detector. Separation was performed on an Agilent ZORBAX Eclipse Plus C18 column (4.6 mm × 250 mm, 5 μm). The mobile phase consisted of water (A) and acetonitrile (B), and the gradient conditions were as follows: 0–5 min, 15% B; 5–30 min, 15–40% B; and 30–50 min, 40–50% B. The column temperature was set at 30 °C, and the flow rate was set at 1.0 mL/min. The detection wavelength was 203 nm, and the injection volume was 20 μL.

### 3.6. Optimization of the Extraction Conditions for Bioactive Steroidal Saponins

The extraction parameters of the bioactive steroidal saponins by the selected NADES were optimized using RSM, which is a statistical technique for studying the interactions between factors in an appropriate variable range. On the basis of preliminary single-factor experiments, the water content (%) in NADESs (A), extraction time (B), and liquid–solid ratios (C) were selected as three independent variables to optimize the extraction parameters by Box–Behnken design (BBD). The total extraction yields of the four bioactive steroidal saponins were taken as the response of the design experiments. Regression analysis was performed according to the experimental data, and additional confirmation experiments were conducted to confirm the validity of the statistical experimental strategies. The Design-Expert Ver. 8.0.6 software (Stat-Ease Inc., Minneapolis, MN, USA) was used for the generation and evaluation of the experimental design.

### 3.7. Microstructure of Plant Material

A JEOL (JEOL Ltd., Tokyo, Japan) JSM-6390A field emission scanning electron microscope in the secondary electron mode with 15-kV accelerating voltage was used for observing the microstructure of the raw and extracted DNR powder. Before imaging, the samples were coated with a Pt layer of approximately 10-nm thickness.

### 3.8. Recovery of Bioactive Steroidal Saponins from NADESs

Recovery of the four steroidal saponins from the NADESs was conducted based on microporous resin column chromatography. A glass column filled with 10 g D101 macroporous resin was used and the bed volume (BV) was 20 mL. The NADESs solution (1 mL) was loaded at the flow rate of 3 BV/h. After the target compounds have been adsorbed in the resin, the column was washed with deionized water (100 mL) first, and then eluted with 100 mL ethanol. Then, the ethanolic fraction was concentrated under vacuum, and analyzed by HPLC as described in Section 3.4. The recoveries of four steroidal saponins were calculated and the above experiments were carried out by triplicate.

## 4. Conclusions

In the present study, a green and efficient extraction method using NADESs as the extraction solvent was established for extraction of bioactive steroidal saponins from DNR. A total of twenty-one types of NADESs were successfully synthesized and used to extract four bioactive steroidal saponins, including protodioscin, protogracillin, pseudoprotodioscin, and pseudoprotogracillin from DNR. The results indicated that the tailor-made NADESs were efficient solvents for the extraction of steroidal saponins, and the ChCl-Mal with a molar ratio of 1:1 proved to be the most efficient NADES. Then, the extraction parameters of ChCl-Mal for extraction of the four bioactive steroidal saponins were optimized using RSM and the optimal extraction condition was water content in NADESs, 40%; extraction time, 20 min; and liquid–solid ratios, 32 mL/g. Moreover, the microstructure of the DNR powder before and after ultrasonic extraction were observed by field emission scanning electron microscope. The recovery of the extracted steroidal saponins from NADESs was using D101 macroporous resin and the recovery yields of protodioscin, protogracillin, pseudoprotodioscin, and pseudoprotogracillin were 79.90%, 68.12%, 67.27%, and 74.8%, respectively. The present research suggests that NADESs are efficient solvents which could be used as green media for extraction of bioactive steroidal saponins from DNR, and the data acquired in this study might contribute to further NADESs application in extraction of bioactive compounds from natural products.

## Figures and Tables

**Figure 1 molecules-26-02079-f001:**
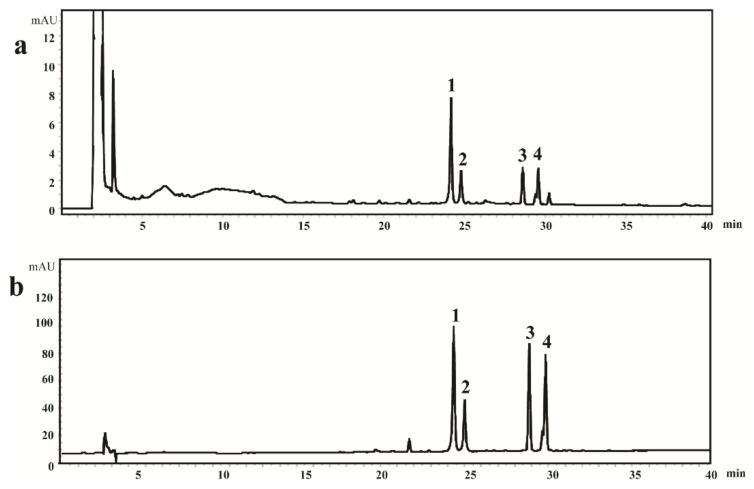
The typical HPLC chromatograms of: (**a**) Dioscoreae Nipponicae Rhizoma sample and (**b**) four steroidal saponins reference standards. (1. protodioscin, 24.15 min; 2. protogracillin, 24.77 min; 3. pseudoprotodioscin, 28.60 min; and 4. Pseudoprotogracillin, 29.56 min).

**Figure 2 molecules-26-02079-f002:**
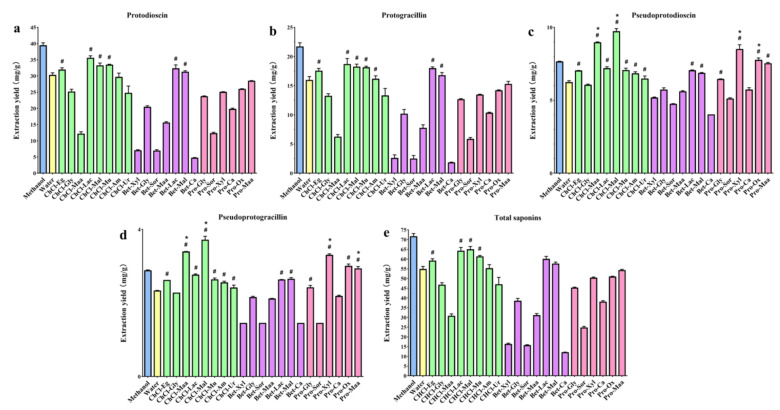
Extraction yields of different solvents for (**a**) protodioscin, (**b**) protogracillin, (**c**) pseudoprotodioscin, (**d**) pseudoprotogracillin, and (**e**) four total steroidal saponins. Error bars indicate the standard deviations (*n* = 3). Extraction yields of NADESs compared with water are indicated with # *p* < 0.05, mean value > water and * *p* < 0.05, mean value > methanol.

**Figure 3 molecules-26-02079-f003:**
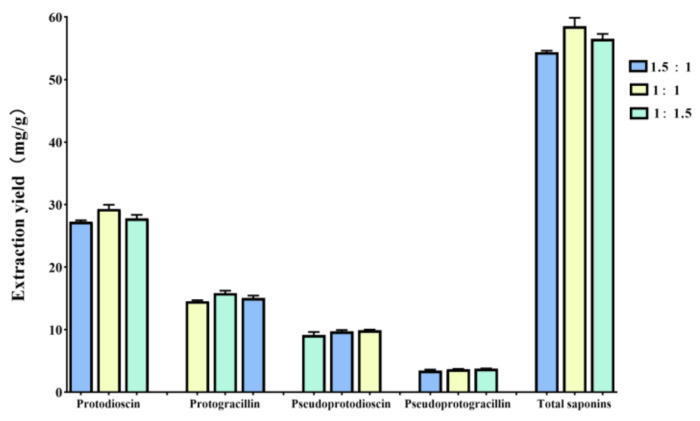
Extraction yields of different molar ratios (1.5:1, 1:1, and 1:1.5) of ChCl-Mal for protodioscin, protogracillin, pseudoprotodioscin, pseudoprotogracillin, and four total steroidal saponins. Error bars indicate the standard deviation (*n* = 3).

**Figure 4 molecules-26-02079-f004:**
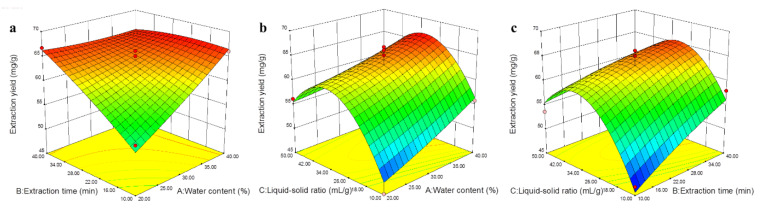
Response surface plots of the model for extraction of steroidal saponins from Dioscoreae Nipponicae Rhizoma (DNR). (**a**) water content (%) and extraction time (min); (**b**) water content (%) and liquid-solid ratio (mL/g); (**c**) liquid-solid ratio (mL/g) and extraction time (min).

**Figure 5 molecules-26-02079-f005:**
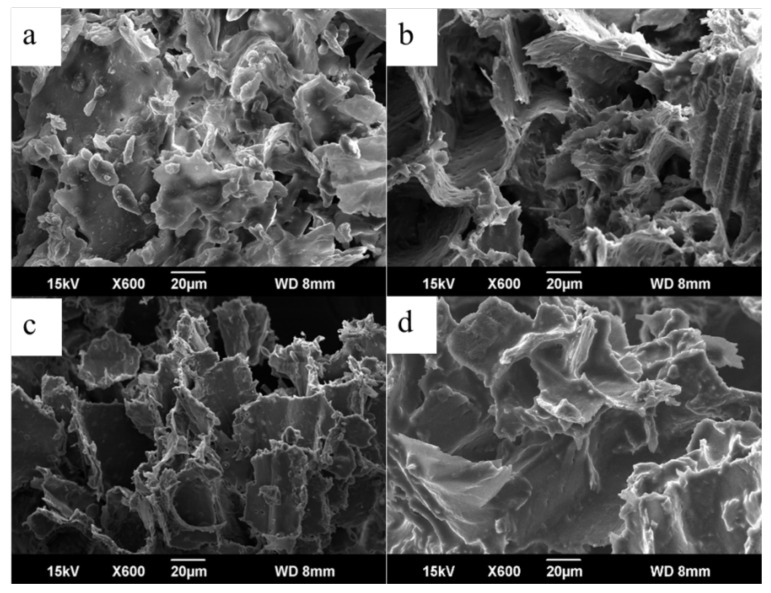
The field emission scanning electron microscope images of DNR powder: (**a**) before ultrasonic extraction and after ultrasonic extraction in (**b**) water, (**c**) methanol, and (**d**) the selected NADES (ChCl-Mal).

**Figure 6 molecules-26-02079-f006:**
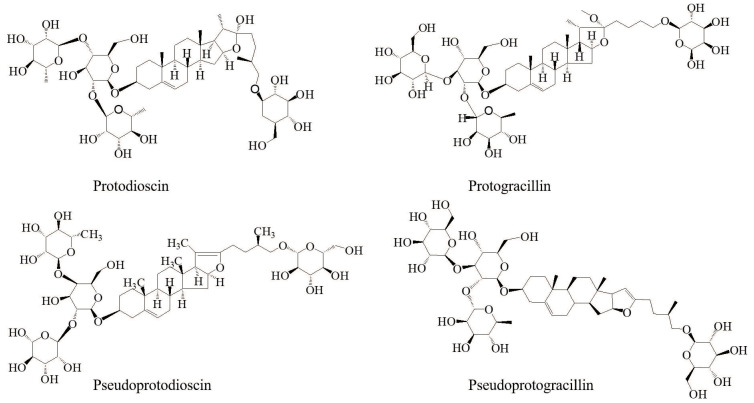
Chemical structures of protodioscin, protogracillin, pseudoprotodioscin, and pseudoprotogracillin.

**Table 1 molecules-26-02079-t001:** Regression equation, limit of detections (LODs), limit of quantifications (LOQs), precision, repeatability, stability, and recovery of four bioactive steroidal saponins.

No.	Analyte	Regression Equation	*r* ^2^	Linear Range (mg/mL)	LODs (μg/mL)	LOQs (μg/mL)	Precision		Repeatability RSD (%)	Stability RSD (%)	Recovery
							Intra-Day RSD (%)	Inter-Day RSD (%)			(mean ± SD, %)
1	protodioscin	Y = 1657.1X + 41.135	0.9993	0.050–0.84	0.054	0.18	1.1	2.2	1.6	1.2	104.7 ± 1.6
2	protogracillin	Y = 1067.8X + 20.476	0.9997	0.030–0.49	0.076	0.26	1.4	2.7	1.1	3.0	104.8 ± 2.4
3	pseudoprotodioscin	Y = 3978.2X − 100.22	0.9992	0.040–0.64	0.048	0.16	1.2	2.1	2.1	2.6	107.5 ±1.9
4	pseudoprotogracillin	Y = 3549.6X − 32.319	0.9997	0.030–0.44	0.037	0.12	1.3	2.4	1.7	1.7	98.8 ± 1.9

**Table 2 molecules-26-02079-t002:** List of natural deep eutectic solvents (NADESs) synthesized in this study.

No.	Abbreviation	Hydrogen Bond Acceptors (HBAs)	Hydrogen Bond Donors (HBDs)	Mole Ratios
1	ChCl-Eg	Choline chloride	Ethylene glycol	1:2
2	ChCl-Gly	Choline chloride	Glycerol	1:2
3	ChCl-Maa	Choline chloride	dl-Malic acid	1:1
4	ChCl-Lac	Choline chloride	Lactic acid	1:1
5	ChCl-Mal	Choline chloride	Malonic acid	1:1
6	ChCl-Mu	Choline chloride	l-Methylurea	1:1
7	ChCl-Am	Choline chloride	Acetamide	1:1
8	ChCl-Ur	Choline chloride	Urea	1:2
9	Bet-Xyl	Betaine	Xylitol	1:1
10	Bet-Gly	Betaine	Glycerol	1:2
11	Bet-Sor	Betaine	d-Sorbitol	1:1
12	Bet-Maa	Betaine	dl-Malic acid	1:1
13	Bet-Lac	Betaine	Lactic acid	1:1
14	Bet-Mal	Betaine	Malonic acid	1:1
15	Bet-Ca	Betaine	Citric acid	2:1
16	Pro-Gly	l-proline	Glycerol	5:2
17	Pro-Sor	l-proline	d-Sorbitol	1:2
18	Pro-Xyl	l-proline	Xylitol	1:1
19	Pro-Ca	l-proline	Citric acid	1:1
20	Pro-Ox	l-proline	Oxalic acid	1:1
21	Pro-Maa	l-proline	dl-Malic acid	1:1

**Table 3 molecules-26-02079-t003:** The experimental orders, levels of variables, and response values in Box–Behnken design.

Run.	Water Content in NADESs (A, %)	Extraction Time (B, min)	Liquid–Solid Ratio (C, mL/g)	Total Extraction Yields of Four Steroidal Saponins (mg/g)
1	40 (1)	25 (0)	50 (1)	61.69
2	30 (0)	40 (1)	10 (−1)	58.15
3	40 (1)	25 (0)	10 (−1)	56.01
4	40 (1)	40 (1)	30 (0)	64.55
5	20 (−1)	10 (−1)	30 (0)	54.62
6	30 (0)	25 (0)	30 (0)	63.25
7	30 (0)	25 (0)	30 (0)	66.21
8	30 (0)	10 (−1)	50 (1)	53.81
9	20 (−1)	40 (1)	30 (0)	66.79
10	20 (−1)	25 (0)	10 (−1)	45.14
11	20 (−1)	25 (0)	50 (1)	56.36
12	30 (0)	25 (0)	30 (0)	63.13
13	30 (0)	25 (0)	30 (0)	58.59
14	40 (1)	10 (−1)	30 (0)	66.13
15	30 (0)	40 (1)	50 (1)	56.28
16	30 (0)	10 (−1)	10 (−1)	46.65
17	30 (0)	25 (0)	30 (0)	65.17

**Table 4 molecules-26-02079-t004:** The analysis of variance (ANOVA) results of the quadratic multiple regression model for steroidal saponins.

Variables	Sum of Squares	df	Mean Square	F-Value	*p*-Value
Model	631.18	9	70.13	8.40	0.0052
A-Water content	81.09	1	81.09	9.72	0.0169
B-Extraction time	75.31	1	75.31	9.02	0.0198
C-Liquid–solid ratio	67.51	1	61.51	7.37	0.0300
AB	47.25	1	47.25	5.66	0.0489
AC	7.65	1	7.65	0.92	0.3703
BC	20.42	1	20.42	2.45	0.1617
A^2^	0.73	1	0.73	0.087	0.7767
B^2^	1.85	1	1.85	0.22	0.6520
C^2^	332.50	1	332.50	39.84	0.0004
Residual	58.42	7	8.35		
Lack of Fit	24.26	3	8.09	0.95	0.4976
R^2^	0.9153				
Adj R^2^	0.8064				

## Data Availability

The data presented in this study are available on request from the corresponding authors.

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
