# Peer review of "Natural Deep Eutectic Solvents for the Extraction of Bioactive Steroidal Saponins from Dioscoreae Nipponicae Rhizoma"

_molecules, 2021, doi:10.3390/molecules26072079_

Round 1

Reviewer 1 Report

This article present the evaluation of Natural Deep Eutectic Solvents (NADESs) for the extraction of steroidal saponins from Dioscoreae Nipponicae Rhizoma  employing ultrasonic extraction of four bioactive steroidal saponins, including pro- 79 todioscin, protogracillin, pseudoprotodioscin and pseudoprotogracillin. The extraction conditions were optimized including the water content of NADESs, extraction time and liquid-solid ratio using response surface methodology (RSM), and field emission scanning electron microscope. Morever the recovery of the four steroidal saponins from selected NADESs was tentatively carried out by macroporous resins.

The paper is well organized and written and brings together important information in this field. However, it has some aspects that need to be reviewed before publication.

HPLC conditions were optimized and the chromatographic separation obtained present an elevated analysis time.

Method validation of quantitative analysis of the four bioactive steroidal saponins in 107 DNR was performed but some questions have been explained in the text:

- ¿Calibration curves are solvent calibration curves?

- ¿Where is the concentration using in study of intra- and inter-day precision of the method?

- In the recovery assays using the equation: Recovery (%) = (Detected amount – Original amount) / Spiked amount × 100% ¿How the detected amount is calculated? In solvent calibration? There isn´t correct by the matrix effect is not correct. It is necessary calculated de matrix effect of the method comparing solvent calibration with matrix-matched calibration and using  matrix-matched calibration for calculate  detected amount.

- The matrix effect (ME) have been  calculated and included in Table 1

The screening of NADESs for extraction of bioactive steroidal saponins have been correctly plateaued and the optimization of the extraction conditions for bioactive steroidal saponins it is a beautiful study by response surface methodology.

The recovery of extracted bioactive compounds from NADESs using D101 macroporous resin column chromatograph have been not optimized and the recoveries obtained are in the range from 79.90% to 67.27%. ¿Why using a D101 macroporous resin?

Another minor questions have been revised in the paper:

-Line 55: eliminate the repeated word Morever

-Revise Table 1: it is in two pages and it is not completely visible

-Table 1: recoveries have been shown as recovery ± sd

-Revise Table 2: it is in two pages 

Author Response

Response to Reviewer 1 Comments

Method validation of quantitative analysis of the four bioactive steroidal saponins in DNR was performed but some questions have been explained in the text:

Point 1: Calibration curves are solvent calibration curves?

Response: The calibration curves of the four steroidal saponins were performed with six different concentrations of working standard solutions in triplicate. The standard solutions were prepared as follows: The reference compounds of protodioscin, protogracillin, pseudoprotodioscin and pseudoprotogracillin were accurately weighed and dissolved in DES (ChCl-Mal), and then diluted eight times with 50% methanol to the concentrations of 840, 490, 640 and 320 μg/mL, respectively. The standard solutions containing four reference standards were prepared by appropriate dilution to a series of proper concentrations. The standard solutions were stored at 4 °C for further HPLC analysis. The “preparation of standard solutions” has been added in Section 3.4 in the revised manuscript.

Point 2: Where is the concentration using in study of intra- and inter-day precision of the method?

Response: The precision of the method was determined by the intra- and inter-day variations. For intra-day test, the same sample was analyzed for six times within the same day, while for inter-day test, the sample was examined in duplicates for consecutive three days. The intra- and inter-day precision of the method has been revised in the manuscript. Thanks for the advice.

Point 3: In the recovery assays using the equation: Recovery (%) = (Detected amount – Original amount) / Spiked amount × 100%. How the detected amount is calculated? In solvent calibration? There isn´t correct by the matrix effect is not correct. It is necessary calculated matrix effect of the method comparing solvent calibration with matrix-matched calibration and using matrix-matched calibration for calculate detected amount. The matrix effect (ME) has been calculated and included in Table 1.

Response: Thanks for the suggestion. For method validation of quantitative analysis of the four bioactive steroidal saponins in DNR, the recovery test was used to evaluate the accuracy. A known amount of the four steroidal saponins reference standard solutions were added into the same samples in sextuplicate, and then the samples were extracted and analyzed with the same procedures as described in Section 3.3 Extraction procedures. The samples were extracted by ultrasonic (300W, 40 kHz) for 20 min at room temperature. After extraction, the mixed suspension was centrifuged at 13000 rpm/min for 10 min. Then, the supernatant was diluted eight times using 50% methanol and filtered through membrane filters (0.22 μm) prior to HPLC analysis. Then the amounts of the four steroidal saponins in the recovery samples were calculated by the calibration curves. The calibration curves of the four steroidal saponins were performed with six different concentrations of standard solutions in triplicate. The standard solutions were prepared as follows: The reference compounds of protodioscin, protogracillin, pseudoprotodioscin and pseudoprotogracillin were accurately weighed and dissolved in DES (ChCl-Mal), and then diluted eight times with 50% methanol to the concentrations of 840, 490, 640 and 320 μg/mL, respectively. The standard solutions containing four reference standards were prepared by appropriate dilution to a series of proper concentrations. So in this present work, we used the matrix-matched calibration curves for calculate the amount of the four steroidal saponins.

Point 4: The recovery of extracted bioactive compounds from NADESs using D101 macroporous resin column chromatography has been not optimized and the recoveries obtained are in the range from 79.90% to 67.27%. Why using a D101 macroporous resin?

Response: The recovery of the four bioactive steroidal saponins from NADESs was tentatively applied by using macroporous resin column chromatography according to our previous research [Duan, L.; Zhang, W.H.; Zhang, Z.H.; Liu, E.H.; Guo, L. Evaluation of natural deep eutectic solvents for the extraction of bioactive flavone C-glycosides from Flos Trollii. Microchem. J. 2019, 145, 180-186.]. The D101 macroporous resin is a common, accessible and inexpensive packing material for column chromatography, hence in this study, the recovery of extracted bioactive compounds from NADESs were using D101 macroporous resin column chromatography.

Point 5: Line 55: eliminate the repeated word Morever.

Response: Sorry for the mistake. The repeated word “However” has been deleted in the revised manuscript.

Point 6: Revise Table 1: it is in two pages and it is not completely visible.

Response: Thanks for the suggestion. The Table 1 has been displayed in one page in the revised manuscript.

Point 7: Table 1: recoveries have been shown as recovery ± sd

Response: Thanks for the suggestions. The recovery in Table1 has been shown as mean ± SD.

Point 8: Revise Table 2: it is in two pages 

Response: Thanks for the suggestion. The Table 2 has been displayed in one page in the revised manuscript.

Reviewer 2 Report

the work is clear and interesting, providing important tools for the validation of extraction methods using green chemical solvents. This methodology and its improvement and validation is perfectly inserted in the objectives of the UN 2030 agenda, which makes it even more important.
I just have to point out a failure on line 55 where a word is repeated.

Author Response

Response to Reviewer 2 Comments

Point 1: The work is clear and interesting, providing important tools for the validation of extraction methods using green chemical solvents. This methodology and its improvement and validation is perfectly inserted in the objectives of the UN 2030 agenda, which makes it even more important. I just have to point out a failure on line 55 where a word is repeated.

Response: Sorry for the mistake. The repeated word “However” in line 55 has been deleted in the revised manuscript.

Reviewer 3 Report

The manuscrip entitled “Natural Deep Eutectic Solvents for the Extraction of Bioactive Steroidal Saponins from Dioscoreae Nipponicae Rhizoma” describes a simple and environmentally friendly ultrasonic extraction method based on NADESs to extract four bioactive steroidal saponins from DNR: protodioscin, protogracillin, pseudoprotodioscin and pseudoprotogracillin. A total of twenty-one types of choline chloride, betaine, and L-proline based NADESs were synthesized for initial screening, and the NADES composed of 1:1 molar ratio of choline chloride and malonate showed best extraction efficiency for the four steroidal saponins compared with other NADESs. The analytical method is very accurate and precise and furnish a very detailed analysis of the different extraction parameters. The manuscript is very well structured, the abstract resume in accurate way all principal part of the work and report the essential information. The introduction is proportionated with the importance of the work and describe in exhaustive way all principles and theoretical concepts used in the text. Results and Discussion part is excellent structured and describes in linear way the work that consist in a preliminary optimization of the extraction conditions considering all principal analytical variables and the comparison of the extraction efficiency of NADES with water and methanol is chemically appreciate. Conclusions are short and concise and resumes all expected values. Furthermore, the English language used in manuscript is of good level but some short modifications are necessary.

For all reason illustrated above, I suggest the publication of this manuscript in Molecules.

If the editor decide for the publication of manuscript, I have minor revisions suggestions for the authors that can be found in attached file. 

Author Response

Response to Reviewer 3 Comments

Point 1: Line 55. Please, remove one "However".

Response: Sorry for the mistake. The repeated word “However” has been deleted in the revised manuscript.

Point 2: Line 67. In this sentence the authors describe the main applications of DESs emphasizing pharmaceuticals, catalysis, electrochemistry and extractions. However, the references included describes very well these applications for pharmaceuticals (ref. 13), electrochemistry (ref. 15) and extractions (ref. 14, 16) but none of them relate to catalysis in an explanatory way. For this reason, in accordance with the authors, I gentle suggest the introduction of the following reference to emphasize the application of DESs in catalysis field: Maiuolo, L.; Algieri, V.; Olivito, F.; De Nino, A. Recent Developments on 1,3-Dipolar Cycloaddition Reactions by Catalysis in Green Solvents. Catalysts, 2020, 10, 65, DOI: 10.3390/catal10010065.

Response: Sorry for the mistake. The reference 17 has been added in the revised manuscript.

Point 3: Line 91-92. Please replace the period with a comma and continue the sentence.

Response: Sorry for the mistake. The sentence has been revised as “In order to achieve a rapid and efficient analysis of the four bioactive steroidal saponins (protodioscin, protogracillin, pseudoprotodioscin and pseudoprotogracillin) in DNR, several HPLC conditions, including mobile phases (water-methanol, water-acetonitrile, formic acid water-methanol and formic acid water-acetonitrile), flow rates (0.9 mL/min, 1.0 mL/min and 1.1 mL/min), and column temperatures (20 °C, 30 °C and 40 °C) were compared and optimized.”

Point 4: Line 99. Please, insert the number of compound after the name.

Response: Thanks for the suggestion. The numbers of the compounds, protodioscin (1), protogracillin (2), pseudoprotodioscin (3) and pseudoprotogracillin (4), have been added in the revised manuscript.

Point 5: Line 102. Please, replace Figure 2 with Figure 1.

Response: Thanks for the suggestion. The Figure 2 has been replaced with Figure 1 in the revised manuscript.

Point 6: Line 105. Please, add the exact Retention Time (RT) for the four substances at performed HPLC conditions.

Response: The retention time of protodioscin (24.15 min), protogracillin (24.77 min), pseudoprotodioscin (28.60 min), pseudoprotogracillin (29.56 min) have been added in the revised manuscript.

Point 7: Please, replace with malonic acid.

Response: Sorry for the mistake. The “malonate acid” and “malonate” have been replaced with “malonic acid” in the revised manuscript.

Point 8: Line 189. Please, replace with compared.

Response: The word has been replaced with “compared” in the revised manuscript.

Point 9: Line 340. Please, replace with extraction.

Response: Sorry for the mistake. The word has been replaced with “extraction” in the revised manuscript.
